A framework for near-real time monitoring of diversity patterns based on indirect remote sensing, with an application in the Brazilian Atlantic rainforest

Paz Andrea paz.andreita@gmail.com 1 2 3
Silva Thiago S. 4 5
Carnaval Ana C. 1 2
1 Department of Biology, City College of New York , New York , NY , United States of America
2 Ph.D Program in Biology, City University of New York, Graduate School and University Center , New York , NY , United States of America
3 Department of Environmental Systems Science, Institute of Integrative Biology, Swiss Federal Institute of Technology, Zurich , Zürich , Switzerland
4 Instituto de Geociências e Ciências Exatas, Departamento de Geografia, Ecosystem Dynamics Observatory, Universidade Estadual Paulista , Rio Claro , São Paulo , Brazil
5 Biological and Environmental Sciences, Faculty of Natural Sciences, University of Stirling , Stirling , United Kingdom
Jaffé Rodolfo
Electronic publication date: 2022 Jun 29
Publication date: 2022
Volume: 10
Electronic Location ID: e13534
Received 2022 Feb 17; Accepted 2022 May 12
Copyright: ©2022 Paz et al.
Copyright year: 2022
Copyright holder: Paz et al.
License: This is an open access article distributed under the terms of the Creative Commons Attribution License, which permits unrestricted use, distribution, reproduction and adaptation in any medium and for any purpose provided that it is properly attributed. For attribution, the original author(s), title, publication source (PeerJ) and either DOI or URL of the article must be cited.
License URL: https://creativecommons.org/licenses/by/4.0/

Keywords: Biodiversity, Prediction, Monitoring, Richness, Phylogenetic diversity

Funding: Fulbright-Colciencias fellowship CCNY Cluster for the study of biodiversity and environmental change seed award Fundação de Amparo à Pesquisa do Estado de São Paulo 2013/50297-0 National Science Foundation DEB 1343578 DEB-1343612 Andrea Paz was funded by a Fulbright-Colciencias fellowship and a CCNY Cluster for the study of biodiversity and environmental change seed award. This study used data collected by a collaborative project co-funded by the Fundação de Amparo à Pesquisa do Estado de São Paulo (FAPESP; 2013/50297-0), the National Science Foundation (NSF; DEB 1343578, DEB-1343612), and NASA. The funders had no role in study design, data collection and analysis, decision to publish, or preparation of the manuscript.

==============================
Monitoring biodiversity change is key to effective conservation policy. While it is difficult to establish in situ biodiversity monitoring programs at broad geographical scales, remote sensing advances allow for near-real time Earth observations that may help with this goal. We combine periodical and freely available remote sensing information describing temperature and precipitation with curated biological information from several groups of animals and plants in the Brazilian Atlantic rainforest to design an indirect remote sensing framework that monitors potential loss and gain of biodiversity in near-real time. Using data from biological collections and information from repeated field inventories, we demonstrate that this framework has the potential to accurately predict trends of biodiversity change for both taxonomic and phylogenetic diversity. The framework identifies areas of potential diversity loss more accurately than areas of species gain, and performs best when applied to broadly distributed groups of animals and plants.

Introduction

Real-time monitoring of biodiversity is a fundamental goal of biological conservation. If implemented at large geographical scales, it can strongly impact policy and action. Yet the difficulty of establishing real- or near-real time diversity monitoring on the ground poses a significant challenge for large-scale conservation planning and management (Scholes et al., 2008; Navarro et al., 2017). The increased availability of free remote sensing data, encompassing broad geographic extents at fine spatial and temporal resolution, makes the promise of real- or near-real time monitoring of biodiversity from space more attainable (Scholes et al., 2012; Turner, 2014; Reddy, 2021). As such, research programs and services that automatically process periodically available satellite images to flag land-use change have begun to take advantage of this potential. The Global Forest Watch and the Amazon DETER initiative (World Resources Institute, 2014; Diniz et al., 2015) are examples of flagship programs that monitor deforestation and forest fires, issuing early alerts for conservation and providing examples of what can now be done in biodiversity monitoring.

Yet satellite imagery can only view so much—and often cannot provide species level identifications or information below the canopy level. Most proposals to measure essential biodiversity variables from remote sensing include only direct measurements such as canopy tree observations and, so far, do not integrate with the genetic dimension of diversity (Reddy, 2021; Skidmore et al., 2021). Even aerial photographs of canopy trees can fail to provide accurate identifications at the species level, particularly in the tropics (Garzon-Lopez et al., 2013). Although new techniques such as hyperspectral satellite imagery show better taxonomic discrimination, they can still only sense canopy plants or lower vegetation in open areas (Papeş et al., 2010). Furthermore, observations of animals are often unfeasible, or limited (but see Fretwell & Trathan (2009) for animal traces that may be detected from satellites in depauperate ecosystems, and Schmaljohann (2020) for an recent example of detection of animal movement with radar data). These limitations raise the question of whether and how species or general diversity patterns that are not directly observable from space may still be monitored remotely.

To overcome these challenges, it has been proposed that organisms such as understory plants and animals may be monitored indirectly, by tracking the environmental conditions that are statistically associated with their presence (Turner et al., 2003; Turner, 2014; Paz et al., 2020). Several studies have identified statistical relationships between metrics that quantify local biodiversity (Rompré et al., 2007; Laurencio & Fitzgerald, 2010; Peters et al., 2016; Zellweger et al., 2016) and environmental variables describing local climates (temperature and precipitation), or landscapes. Not only has the number of biological species (henceforth referred to as species richness) been related to climate and habitat configuration, but so have metrics that reflect the diversity of evolutionary lineages present in a given region, also known as phylogenetic diversity. Phylogenetic diversity combines information about which species are in a given region and how closely or distantly related they are (Faith, 1992). Measuring both dimensions of diversity allows for the identification of regions of mismatch between species richness and phylogenetic diversity, which indicate that the assembly of the local community was not random with respect to the species pool. For instance, when a given environment selects for species that are closely related to each other and therefore share crucial adaptations for that area (also referred to as environmental filtering, Webb et al., 2002), phylogenetic diversity will likely be lower than expected given the number of species. In contrast, whenever inter-specific competition is prevalent, species that are closely related to each other and fulfill similar ecological roles may not be able to co-exist; under these conditions, local species may be more distantly related than expected, and phylogenetic diversity would be high relative to species richness (Webb et al., 2002). Sometimes, however, these mismatches can be misleading: when distantly related species share similar traits due to convergence (as opposed to common ancestry), then environmental filtering may be associated with phylogenetic overdispersion instead of clustering. Moreover, when competitive abilities are very different, then competition can lead to exclusion of species of lower competitive ability and thus only presence of species of the higher competitive ability with similar traits (Cavender-Bares et al., 2009; Mayfield & Levine, 2010).

Because many of the environmental variables relevant to models of diversity patterns under a static environment can be derived from remote sensing data, it has been proposed that near-real time indirect remote sensing may provide a lens to describe and predict the potential displacement of biodiversity in response to continuous environmental shifts (Paz et al., 2021; Turner et al., 2003). In other words, documentation of changes in environmental conditions via remote sensing may allow for the dynamic monitoring of changes of biodiversity components that cannot be directly observed. However, a platform that applies these concepts and allows governments and scientists to envision how ongoing environmental changes may potentially change the distribution of diversity, in near-real-time, is missing. Importantly, it remains to be uncovered how reliably this approach can generate predictions across different geographical areas, and for biological groups of varying levels of diversity, ecological constraints, and geographical range.

Here, we build and test such a platform for near-real time prediction of diversity change. By applying it to environmental and biodiversity data from the Brazilian Atlantic Forest hotspot, we critically evaluate its limitations and the conditions under which it may prove most useful. We focus on the Atlantic Forest to build on our previous studies (Paz et al., 2020; Paz et al., 2021), which demonstrated that climate variables can accurately predict local species richness and phylogenetic diversity at the community level. Here, we apply this new modeling framework to those two measures of biodiversity in the Atlantic Forest.

This new model for biodiversity monitoring through indirect remote sensing is built and implemented in a reproducible machine-learning framework that integrates Google Earth Engine (Gorelick et al., 2017), the R software (R Core Team, 2022), and data describing the diversity of nine taxonomic groups in the Atlantic forest known to differ in their geographic spread and sampling intensity. The input data include per-pixel descriptors of species richness and phylogenetic diversity of five groups of plants and four groups of animals. All groups have been well sampled throughout their range, adequately geo-referenced, and correctly identified by experts (Paz et al., 2021). By gathering and processing remote sensing data to describe local environmental conditions, and by integrating them with the input biological data (here, maps of species richness and phylogenetic diversity), the modeling framework allows the user to (i) describe potential changes in the distribution of species richness and phylogenetic diversity in near-real time, and (ii) predict potential trends in biodiversity change given changes in environments. These trends are predicted only within the region of model training, and within the 2002–2019 time interval. As such, it is here limited to the Atlantic Forest of Brazil, and for that period of time.

Because machine learning techniques are known to need a substantial amount of data (Van Der Ploeg, Austin & Steyerberg, 2014; Kuhn & Johnson, 2016), and given that species with smaller ranges might be more prone to be affected by history or chance (Paz et al., 2020), we expect that those groups of organisms that have broad geographical ranges, and thus a high number of observations to train the model from, will be best suited for this modeling exercise. As such, we first apply the model to nine groups that differ in their geographical spread, and evaluate how sensitive it is to these distinct geographical ranges. Then, to provide a more stringent test of the modeling platform, we verify how accurately the model can retroactively predict changes in species richness that have been documented in the Atlantic Forest in decades past. Specifically, we use the model to reconstruct potential species loss and gain in two groups of organisms known to be closely impacted by climate, and which have been historically inventoried in this region. To that end, we use two databases that gather inventory data on species presence (but not phylogenetic diversity): ATLANTIC-epiphytes (Ramos et al., 2019) and the ATLANTIC-amphibians database (Vancine et al., 2018).

Methods

To illustrate our modeling framework and evaluate its sensitivity to the geographical range of the group being modeled, we first applied it to nine organismal groups that differ in their spatial distribution. To this end, for each group, we combined its species richness map with temporally-explicit environmental data (temperature and precipitation), derived from remote sensing sources, to generate a model that describes how they are associated. We then performed the same procedure with the group’s phylogenetic diversity map, to model this second biodiversity metric as well. We evaluated model fit for all groups and for both diversity metrics, exploring how they relate to the total range size of the group and the median individual species range size. Models with the best fit were then projected over time, yearly, based on that year’s environmental conditions. Those yearly projections allowed us to map trends in diversity change across time (per-pixel loss or gain of species richness, and per-pixel loss or gain of phylogenetic diversity), for each group. They also allowed us to map areas of potential mismatch between species richness and phylogenetic diversity—which often indicate important ecological processes—and identify whether they may be changing over time. Finally, we applied our framework to an available dataset of observations of amphibians and epiphytes over time, to test the accuracy of the model predictions.

a. Biological datasets and maps of species richness and phylogenetic diversity

To illustrate our biodiversity monitoring model and evaluate its sensitivity to the geographical range of the group, we first applied it to nine groups that differ in their spatial distribution (five groups of plants, and four groups of animals; Table 1). The plants included a group within the Bromelioideae subfamily (Aguirre-Santoro, Stevenson & Michelangeli, 2016; Aguirre-Santoro, 2017; Brown et al., 2020), three groups in the family Melastomataceae [in the Miconieae tribe (Goldenberg et al., 2008; Michelangeli et al., 2008; Caddah, 2013; Reginato & Michelangeli, 2016; Brown et al., 2020), in the genus Bertolonia (Bacci et al., 2020; Paz et al., 2021), and in the tribe Cambessedesieae (Bochorny et al., 2019; Paz et al., 2021)], and a group including Fridericia and allies (Kaehler, Michelangeli & Lohmann, 2019; Paz et al., 2021), in the plant family Bignoniaceae. The animals included the clearwing butterflies that compose the tribe Ithomiini (Brown et al., 2020), treefrogs of the genus Boana (Vasconcelos et al., 2014; Brown et al., 2020), horned frogs of the genus Proceratophrys (Brown et al., 2020), and a group of birds in the subfamily Thraupinae (Burns et al., 2014). These groups and their species differ in their distributions: some have small individual species distributions and small total geographical spread (like the Bromelioideae plants and the Boana frogs), others have broad individual species distributions and broad group-wide spatial coverage (like the Thraupinae), while others have discordant patterns (like the the Fridericia and allies, with small individual ranges but a broad group coverage; see Table 1, sorted from broader geographic distribution to smallest).

Table 1 Biological data used for modeling.

Biological data used to model species richness and phylogenetic diversity (five plant groups and four animal groups). Table includes the median area of distribution of species within each group, and the total area of distribution of each group. Data from Paz et al. (2021).

Clade	Taxonomic level	Median individual area of distribution (km2)	Total area of distribution for the group in the AF (km2)	
Thraupinae	Subfamily	680,022	1,330,613	
Fridericia & allies	Group	85	1,282,733	
Miconieae	Tribe	44,486	1,197,890	
Ithomiini	Tribe	321,331	1,147,465	
Boana	Genus	240	824,368	
Proceratophrys	Genus	3,266	766,577	
Cambessedesieae	Tribe	1,887	755,871	
Bertolonia	Genus	1,970	240,376	
Bromelioideae	Subfamily	41	195,639	

For each group, a map of species richness was downloaded from the Dryad repository of Paz et al. (2021); here, we provide a brief description of the methods used in that publication. Those maps were originally based on individual species distribution maps developed at a 10 km resolution (Paz et al., 2021), and created through the superimposition of alpha hulls representing the ranges of each species. Alpha hulls are a generalization of convex hulls, allowing for more shape flexibility, with lines that are allowed to curve and match the point distributions more closely (Meyer, Diniz-Filho & Lohmann, 2017). Once alpha hulls representing the ranges of each species in group were obtained, they were stacked and summed to obtain a species richness map, which was used in the present analysis (Paz et al., 2021). As any outline-based maps, alpha hulls may overpredict individual species ranges; however, they have been shown to do so less often than other commonly used techniques, such as minimum convex. Although they may lead to overprediction of species richness, this issue is more prevalent at smaller resolutions (Graham & Hijmans, 2006; Hurlbert & Jetz, 2007).

For each group, a map of phylogenetic diversity, as measured with Faith’s PD (Faith, 1992), was also downloaded from Paz et al. (2021). To obtain that map of the spatial distribution of phylogenetic diversity, the species richness map and community composition information of each group were previously combined with phylogenetic information describing the evolutionary relationships among all species in the group, using the software Biodiverse (Laffan, Lubarsky & Rosauer, 2010) as described in Paz et al. (2021).

b. Environmental descriptors

Environmental data to train and project the near-real time models were obtained from remote sensing sources and used to create bioclimatic layers describing the conditions experienced at different time windows. For that, we collected Normalized Difference Vegetation Index (NDVI) data and temperature data from NASA’s Moderate Resolution Imaging Spectroradiometer (MODIS; Vermote & Wolfe, 2015) instrument, and precipitation data from the Climate Hazards Infrared Precipitation with Stations (CHIRPS; Funk et al., 2015) dataset. The MODIS data products are available bi-weekly since 2002, at a minimum resolution of 500m and 1km respectively; CHIRPS data are available monthly between 1981 and the present, at a minimum resolution of 0.05 degrees (∼5.5 km). Using a custom-created python script for Google Earth Engine (GEE, Gorelick et al., 2017), we obtained all NDVI and temperature MODIS data and CHIRPS precipitation data representing conditions experienced in the Atlantic Forest from 2002 to the present, using the forest shapefile from Paz et al. (2021). In GEE, we summarized climate data as monthly means, maximum and minimum values and NDVI data as yearly medians at a 10 km resolution. All summary layers were downloaded into R for further processing and for model building. Based on those monthly data, sets of 19 bioclimatic layers representing summaries of temperature and precipitation (Table 2) were built using the bioclim function of the dismo package (Hijmans et al., 2011). To calibrate the machine learning model, we used built bioclimatic and NDVI layers for the 2002–2014 time period; for projections of the model, we used yearly layers from 2015 to 2019. The python script to summarize and download data from GEE, as well as the R scripts used to generate the bioclimatic layers, are available on the Github repository of this manuscript (https://github.com/andrepazv/NearRealTime). Before proceeding to modeling, we removed all highly redundant (correlated) variables (VIF > 10) using the vifstep function of the usdm R package (Naimi et al., 2014). This reduced the environmental dataset to ten variables that were used in all further analyses (Table 2).

Table 2 Variables used in the machine learning modeling and predictions.

A total of 19 bioclimatic variables were built for training and projecting periods using the dismo package in R. Yearly NDVI values were also included. Monthly temperature and precipitation data were obtained from remote sensing sources as described in the text. After removing all highly correlated variables, the following ten variables were used for model building.

Variable Name	Description	
NDVI	Normalized difference vegetation index	
bio2	Mean Diurnal Range (Mean of monthly (max temp–min temp))	
bio3	Isothermality	
bio4	Temperature Seasonality (standard deviation *100)	
bio8	Mean Temperature of Wettest Quarter	
bio9	Mean Temperature of Driest Quarter	
bio13	Precipitation of Wettest Month	
bio14	Precipitation of Driest Month	
bio18	Precipitation of Warmest Quarter	
bio19	Precipitation of Coldest Quarter	

c. The machine learning model and time projections

To predict the spatial distribution of species richness and phylogenetic diversity as a function of environmental change in near-real time, we independently modeled species richness and phylogenetic diversity with data describing temperature and precipitation. Those models with the best fit were then projected over time, yearly, based on each year’s environmental conditions.

We first trained an ensemble machine learning model to describe species richness, and an ensemble machine learning model to describe phylogenetic diversity, for each of the nine groups (Table 1). For that, we used bioclimatic layers built from MODIS and CHIRPS and NDVI estimates (Table 2) summarized for the 2002–2014 time period. To improve model performance, we performed an ensemble of four machine learning algorithms: Random Forests (rf from Liaw & Wiener, 2002), Neural Networks (nnet from Venables & Ripley, 2002), support vector machines (svmRadial from Karatzoglou et al., 2004), and generalized linear models. These algorithms allow the prediction of continuous variables (as a form of regression), but the first three make no assumption of linearity in the relationship between predictor and response variables, thus allowing for more complex and biologically relevant relationships. Seeking good transferability across time projections, we used a spatial partition method for model tuning and validation (Radosavljevic & Anderson, 2014): for each model, we partitioned the biodiversity data (pixels) into geographic regions, instead of randomly. To do that, for each biodiversity metric (species richness or phylogenetic diversity) and taxon (each one of the nine groups of organisms), we created four spatial partitions using an adapted version of the function get.block in the package ENMeval, partitioning the data into bins across latitude and longitude lines (provided with the full script on GitHub; Muscarella et al., 2014). For each one of the four machine learning algorithms, we employed k-fold cross validation, building k = 4 models, each leaving out one different spatial partition that was used for model validation. Models were trained and tuned using the caretList function in the caretEnsemble R package, with partitions passed through the trainControl argument (Deane-Mayer & Knowles, 2019). Individual tuned models were selected using the root mean square error (RMSE) metric. Using the settings determined as optimal in the validation stage, we then built a full model with all data (full model from hereon). Finally, an ensemble model was built for each organismal group and biodiversity metric, based on a linear combination of the models for all four algorithms, weighted according to their RMSE values. For that, we used the caretEnsemble R package and repeated 10-fold cross-validation (Deane-Mayer & Knowles, 2019). The script used for modeling is available in the GitHub repository for this manuscript.

These models allow for predictions of the response variable (here, species richness and phylogenetic diversity) to be transferred to a subsequent time period based on the same predictor variables (the bioclimatic variables + NDVI, Table 2, Fig. S4). To obtain an estimate of trends in biodiversity change over time, we projected the models in time over five different years, generating species richness and phylogenetic diversity projections for 2015, 2016, 2017, 2018 and 2019. For this, we applied the corresponding model to each year’s predictor variables. Using the model and the five projections over time (from 2015 to 2019), we stacked the maps together and computed the slope of a linear regression for each pixel to estimate trends in biodiversity change for each group and biodiversity metric. To obtain a spatial estimate of the expected mismatch between the biodiversity metrics, we computed the regression of phylogenetic diversity onto species richness for each year, and mapped the residuals of the regression for each year.

d. Ground truthing the modeling framework

We then used another dataset to ground-truth the model framework, contrasting the biodiversity change trends inferred by the model against trends documented in the field. For that, we used information available for two other groups of organisms—epiphytes and amphibians—for which survey data are available across multiple years and across a sufficient percentage of their ranges, using the ATLANTIC-epiphytes (Ramos et al., 2019) and ATLANTIC-amphibian datasets (Vancine et al., 2018). These datasets were made possible by a multi-institutional effort that gathered inventory data for different plant and animal groups, collected across several decades along the Atlantic Forest. Although geographical sampling along the years is sparse, these two datasets have survey information that spans most of the latitudinal extent of the Atlantic Forest and includes contemporary sampling that partly matches the available satellite information. The epiphyte dataset was composed of 89,270 observations from 1824 to 2018 (Ramos et al., 2019). That of the amphibians was composed of 17,619 records, with data from 1940 up to 2015 (Vancine et al., 2018). For both datasets, we created maps of species richness for both datasets at a resolution of 100 km. This resulted in maps of 188 and 137 pixels, respectively, with species richness per pixel varying from 1–365 epiphyte species and from 1–80 amphibian species. Because the ATLANTIC datasets do not include phylogenetic data, this analysis focused solely on species richness and not phylogenetic diversity.

For the purposes of ground-truthing the model, we limited our analyses to the years for which the datasets (i) documented the year of the observations, (ii) provided coordinates for the observation localities and (iii) provided a relatively broad sampling coverage of the region (i.e., sampled sites were not clustered geographically). This resulted in the use of data collected up to the year of 2015 for the amphibians, and up to the year of 2018 for the epiphytes. We then used the available inventory data from all but the last four years of observations to generate maps of species richness (up to the year of 2011 for the amphibians, and 2014 for the epiphytes), measured as the number of species present in each pixel in each year. Following the same modeling procedure applied previously, we combined those species richness maps with bioclimatic variables derived from remote sensing sources as described above, and for the same period, to train a predictive model. The model was then projected onto the four following (most recent) years, for which observational data are available for ground-truthing. As such, we obtained yearly model projections of amphibian species richness for the years of 2012, 2013, 2014, and 2015, and yearly model projections of epiphyte species richness for the years of 2015, 2016, 2017, and 2018. As done previously, we used these yearly model projections to create maps of inferred trends in species richness across time for each of the target groups, computing the slope of a linear regression for each group. We then compared these modelled trends with the observations recorded in the datasets, evaluating both directionality and magnitude of change. To evaluate if results were better than random predictions, we created 10,000 random predictions of gain and loss across the forest, and, for each one, computed the number of rightly predicted pixels. We used those results to create a null distribution of random predictions and compared the observed number of correct predictions with the null distribution. A script used to run the entire framework (in R and python) is available in the GitHub repository for this manuscript.

Results

Model applicability and sensitivity to range size

The analysis of the first nine groups shows that prediction of biodiversity metrics from environmental variables describing temperature, precipitation and canopy cover during the years of 2002–2014 varied substantially in fit across datasets (species richness: R2 values 0.18–0.83; phylogenetic diversity: 0.08–0.74; Table 3). In most cases, the fit of the ensemble model was better than that of any individual (tuned) model, with the fit of each individual model being similar across algorithms. In a few exceptions, however, the fit of the individual model was poor—in those cases, the fit of the ensemble model was even worse (Table S1).

Table 3 R2 of predictive models of species richness and phylogenetic diversity, based on environmental data, developed for nine target taxonomic groups.

Clade	R2 of predictive model for species richness	R2 of predictive model for phylogenetic diversity	
Fridericia & allies	0.83	0.71	
Thraupinae	0.72	0.68	
Miconieae	0.71	0.74	
Ithomiini	0.56	0.59	
Bromelioideae	0.55	0.49	
Boana	0.53	0.52	
Cambessedesieae	0.47	0.34	
Proceratophrys	0.29	0.08	
Bertolonia	0.18	0.46	

Model fit increased with the total area of distribution of the group, when predicting both species richness and phylogenetic diversity (Fig. 1A). However, median individual species range does not appear to be related to model fit (Fig. 1B). Five out of the nine groups had well-fit models for both diversity measures (R2 >  0.5): they included two plant groups (Miconieae and Fridericia & allies) and three animal groups (Thraupinae birds, Ithomiini butterflies and Boana frogs, Fig. 1, Table 3). These three groups are known for having the largest total geographical range in the Atlantic Forest (1,197,890 km2, 1,282,733 km2, 1,330,613 km2, 1,147,465 km2 and, 824,368 km2 respectively), but they have variable individual species ranges (median individual species ranges 44,486 km2, 85 km2, 680,022 km2, 321,331 km2 and 240 km2 respectively, Table 2). Models of all four remaining groups had an R2 smaller than 0.49 for at least one of the two diversity measures (Table 3). All four groups had comparatively small total ranges in the forest, but variable individual ranges (Table 2).

Figure 1 Relationship between model fit and total and individual distributions.

(A) Relationship between the total distribution area (in km2) and predictive model fit (R2) for species richness (white circles) and phylogenetic diversity (black triangles), for nine target groups. (B) Relationship between the median area of distribution of individual species (log transformed km2) and predictive model fit (R2) for species richness (white circles) and phylogenetic diversity (black triangles). Distribution extent was obtained from the species richness maps, R2 values were obtained from the machine learning predictive model.

In the five groups for which the models were able to predict species richness with good accuracy (Miconieae, the Fridericia & allies, Thraupinae birds, Ithomiini butterflies and, Boana frogs; species richness R2 values of 0.71, 0.83, 0.72, 0.56 and, 0.53 respectively; Table 3), the most important predictor variables were all related to precipitation. In Miconieae and the birds, precipitation of the warmest quarter (bio 18) stood up (explaining 25% and 18% of the variation respectively, Table S2). For Fridericia and the Ithomiini butterflies, precipitation of the wettest month (bio 13) was most important variable (explaining 18% of the variation). Finally, for the Boana frogs, precipitation of the driest month (bio 14) was most important (22%, Table S2). Patterns of phylogenetic diversity in these groups were similarly well correlated with local environments (R2 values of 0.74, 0.71, 0.68, 0.59 and, 0.52 respectively) and, again, the most important predictor variables were all related to precipitation. Precipitation of the warmest quarter (bio 18) was the main predictor of phylogenetic diversity in the Miconieae and the Boana frogs (25% and 22% respectively), whereas precipitation of the wettest month was the main driver for the other three groups (15% for the Fridericia, 21% for the Thraupinae and 17% for the Ithomiini, Table S2).

Projections in the five groups with good model fit show similarities and differences in the predicted trends of change in species richness and phylogenetic diversity from 2015 to 2019 (clusters of gain (red) and loss (blue) of predicted diversity across the forest; Fig. 2). In all groups, and for both diversity metrics, a trend toward diversity loss over time is identified in the central region of the forest, in the north of the São Paulo subtropical gap located between the coastal mountains (Amaral et al., 2018; Thom et al., 2020), and into the inland state of Minas Gerais. This is particularly strong in Fridericia & allies (Fig. 2, second row). On the other hand, a trend toward diversity gain (both in species richness and phylogenetic diversity) is detected along the coast, stretching northward from the state of Rio de Janeiro to Espírito Santo (Fig. 2). Models of all groups also detect trends of increasing diversity for both biodiversity metrics in the southern range of the forest (Fig. 2). Maps of the yearly predictions are available in the supplemental material (Figs. S1 and S2).

Figure 2 Models of temporal trends in species richness and phylogenetic diversity between the years of 2015 and 2019 for the five groups with the highest model predictive ability.

Models of temporal trends in species richness (left) and phylogenetic diversity (right) between the years of 2015 and 2019 for the five groups with the highest model predictive ability (R2 > 0.5), from top to bottom: Miconieae, Fridericia & allies, Thraupinae birds, Ithomiinae butterflies and Boana frogs. Maps depict the slope of the regression between the model built with pre-2015 data and the yearly predictions of species richness and phylogenetic diversity between 2015 and 2019 using the framework presented in the paper. Yellow areas depict no change, red areas represent gain of species or phylogenetic diversity, and blue areas represent loss of species or phylogenetic diversity. Darker shades represent larger regression slopes (more gain or loss). For species richness, the slope represents the number of species gained or lost per year; for phylogenetic diversity, the slope represents phylogenetic diversity gained or lost per year. Color scale for all maps is a quantile classification with 20 classes. For species richness, the slopes vary between −9.08 and 9.9 in the Miconieae, between −1.85 and 1.82 in the Fridericia & allies, between −1.09 to 1.55 in the Thraupinae, between −2.64 and 3.04 in the Ithomiinae and between −0.49 and 0.50 in the Boana frogs. For phylogenetic diversity, the slope varies between −33.41 and 43.71 in the Miconieae, between −0.016 and 0.010 in the Fridericia & allies, between -15.50 and 21.70 in the Thraupinae birds, between −0.12 and 0.13 in the Ithomiini butterflies, and between −0.085 and 0.11 in the Boana frogs.

For all five groups, the predicted distribution of the residuals of the regression of phylogenetic diversity and species richness varies across space, and, to a lesser degree, over time (Fig. 3). In the Miconieae, the models suggest that more phylogenetic diversity is expected to accumulate given species richness (positive residuals expanding across time; Fig. 3, top row green colors). However, two areas of lower phylogenetic diversity are flagged over time: one to the northwest, where the deciduous forest is present (Peres et al., 2020), and a second one, along the coast, in the Serra do Mar region (Fig. 3, top row brown colors). In Fridericia & allies, the projections mostly suggest that phylogenetic diversity is expected to accumulate relative to species richness except for sites in the northwest (similar to Miconieae) and a large region in interior São Paulo (Fig. 3, second row, brown colors). The latter is the same region where both Miconiaeae and Fridericia & allies are inferred to undergo an increase in both diversity metrics over time. The animal groups (Thraupinae birds, Ithomiini butterflies and Boana frogs) differ from the two plant groups in the predicted pace of the changes; they appear less drastic over time (Fig. 3). The bird and frog models detect a large region accumulating more phylogenetic diversity than expected in the south of the forest, and suggest this trend to be mostly stable across time except for 2019 (Fig. 3, third and last row, green colors). These two groups also show a consistent prediction of lower phylogenetic diversity than expected in the north of the forest and a somewhat consistent trend in interior Sao Paulo (Fig. 3, third and last row, brown colors). For the butterflies, an opposite pattern emerges—with the south consistently predicted to have less butterfly phylogenetic diversity than expected, and the north having more phylogenetic diversity than expected (Fig. 3, fourth row, brown and green colors respectively).

Figure 3 Modelled change in residuals of the regression of phylogenetic diversity on species richness across time between 2015 and 2019.

From top to bottom: for the Miconieae, the Fridericia & allies, the Thraupinae birds, the Ithomiini butterflies, and the Boana frogs. Green colors represent positive residuals; brown colors negative residuals. Color intensity reflects the magnitude of the residuals (darker colors depicting larger residuals).

Lessons from model testing

When the model framework is applied to inventory data that include multiple records per year (enabled by the ATLANTIC-amphibian and ATLANTIC-epiphytes datasets), this framework predicts the directionality of biodiversity change well. Although the predictive power of the amphibian and epiphyte models is lower than those observed for the groups analyzed in the first part of the study (R2 of 0.39 for epiphytes, 0.21 for amphibians), the modelled trends of potential change in species richness are broadly congruent with those measured with empirical data, particularly regarding species loss (Fig. 4). A comparison of the modelled vs. observed trends in gain or loss of species richness indicates that the model framework correctly predicted the direction of change in 59% of the pixels (52/88 pixels, p-value = 0.08) in the epiphyte study, and 76% of the pixels (28/37 pixels p-value = 0.0008) in the amphibian group (Fig. 4). Even though richness was described using larger pixels in the ATLANTIC datasets relative to the nine other groups (100 km instead of 10 km), not all pixels contained enough information to compute observed trends; as such, fewer pixels are available for comparisons (88 for the epiphytes, 37 for the amphibians).

Figure 4 A comparison of the trends of change in species richness for two ATLANTIC datasets (top: amphibians, bottom: epiphytes).

Maps show the slope of the regression between a training dataset (pooled records pre-2012 for amphibians, and pre-2015 for epiphytes) and the observed (left) and modeled (right) species richness in four subsequent years after that. Blue color depicts areas experiencing loss of species over time, whereas red denotes areas experiencing gain of species over time; gray denotes areas with no data. Darker shades denote larger slopes (stronger gain or loss). The last column shows the mismatch between the observed and predicted direction on of change, with black pixels being correctly predicted and dark grey pixels incorrectly predicted.

These modelled trends of change differ between the two groups and are spatially heterogeneous, with a tendency towards loss of species. A total of 46/88 pixels in the epiphytes model and 20/37 pixels in the amphibian model have a negative slope. The same tendency towards species loss is present in the observed data, although more strongly, especially in the epiphytes (79/88 pixels in the observed epiphyte data, and 26/37 pixels in the amphibian data; Fig. 4). While the predictive models of the epiphytes generally matched the observed trends of species loss, the magnitude of loss was generally underestimated; observed slopes ranged from -104 to 19, while modelled slopes varied from −75 to 28 (Fig. 4, bottom). In the amphibians, the patterns of biodiversity inferred by the model matched the direction of change reasonably well, with a similar estimated strength of change (observed slopes ranging from −5.62 to 5; modelled slopes spanning from −6.7 to 5.5, Fig. 4 top).

Discussion

We propose a novel framework for indirect biodiversity monitoring based on correlations with remotely sensed environmental data, and find that it performs well in some, but not all groups of organisms for which data are available in the Brazilian Atlantic Forest. When applied to groups that are widely distributed—and for which abundant data are available for model training—the model performs very well, statistically. This is the case of the Miconieae, Fridericia & allies, the Thraupinae birds and, the Ithomiini butterflies. However, the ensemble model’s predictive power decreases when applied to other groups, a result also observed when individual models are built (Table S1). The groups for which the models underperform are more spatially restricted, both in terms of overall group range, or, sometimes, including smaller individual species ranges (Table 3). This result is not surprising, as the performance of machine learning methods is expected to improve with the addition of more data (Kuhn & Johnson, 2016). Furthermore, the geographical partition method used for model validation also yields improved performance with better geographical coverage. It is possible that the use of environmental data at a resolution of 10 km may be inappropriate for those groups known to have more restricted distributions, and that those models may improve with environmental data at finer resolutions. Attempts to clarify this issue should nonetheless acknowledge that estimating species distributions at finer resolutions often results in an overestimation of species distributions (Graham & Hijmans, 2006; Hurlbert & Jetz, 2007).

Model outcomes of some groups, including the highly performing models for the Boana frogs, suggest that elements other than total group range are important for model performance. Models of the Boana tree frogs, which have small species ranges and a small total range in the forest, performed well (R2 of 0.53 for species richness, 0.52 for phylogenetic diversity). Models for the two groups with the smallest total area, the Bertolonia and the Bromelioidae (the last one known for having species with the smallest ranges), had comparatively good performance for at least one of the two models (R2 of 0.18 and 0.55 respectively for species richness, 0.46 and 0.49 for phylogenetic diversity). Although the small sample size precludes statistical evaluations (N = 3 groups), it is possible that these results reflect the impact of the ecology of these species, particularly their dependence on the environment, on model performance. Because the ecology and life history traits of treefrogs and bromeliads render them strongly dependent on the abiotic environment, we hypothesize that the models of these groups fit better than expected given their geographical ranges and those of their species. As such, we expect that this model framework will work best for groups of species that are tightly associated with specific environmental conditions (rather than landscape and historical factors), or broadly distributed in a biome of interest.

The generally lower predictive performance of the models, when applied to smaller-ranged groups (R2 ranging from 0.08–0.55, Table 3), is consistent with the outcomes of the models of the ATLANTIC-amphibians and ATLANTIC-epiphyte datasets. Because the latter had fewer pixels (observations), and hence a sparser geographical coverage, lower predictive performance was expected—and detected. Yearly coverage did not appear to dictate whether the direction of change predicted at a site was correctly or incorrectly inferred. Yet, despite these limitations, a comparison between model outputs and the empirical data demonstrates that the proposed framework can perform reasonably well when predicting the directionality of change in species richness—particularly when predicting species loss. This result is not surprising given that this study is evaluating near-real time changes in diversity (i.e., over the span of a few years), when species gain might be more difficult to document via inventory work, and when species gain may lag behind environmental change (Zanatta et al., 2020). On the one hand, even if areas are gaining species, it might take time for populations to become established in enough numbers (or grow, in the case of plants) to be detected by in situ monitoring (Alexander et al., 2018). On the other hand, loss can happen faster in the face of extreme environmental conditions either from local population extinctions, or though movement of the species (Davis & Shaw, 2001; but extinction can also lag in long-lived species or under continued recruitment from seed banks or nearby populations; Alexander et al., 2018). In either case, we argue that the early detection of areas threatened with diversity loss is key to the survival of many species and a useful complement to the early detection of individual species at risk (Stanton et al., 2015).

Although some trends in diversity change are group specific, the models detect certain geographical regions of the forest—e.g., coastal Espírito Santo—that are consistently predicted to gain diversity over time, irrespectively of biodiversity metric. This area is nonetheless under intense anthropogenic disturbance (Tabarelli et al., 2010), and habitat loss might prevent the species accumulation predicted by climate only. In the sense that this model framework does not incorporate land use and land use change, it is important to keep in mind that it so far only provides insight based on macroclimatic shifts. In contrast, the northwestern portion of the forest is consistently predicted to be losing both species and phylogenetic diversity in all groups but the butterflies (loss is only predicted in the northernmost area), and losing more phylogenetic diversity relative to species richness—thus potentially through the loss of species that are distantly related. While these conclusions are suggested by models of the community, they can be tested with species-specific models and targeted inventories.

Likewise, the models may be used to identify potential ecological processes at play—and test them. While patterns of predicted diversity change seem similar between species richness and phylogenetic diversity for all groups, the predicted residuals for each year show that their mismatch varies in time. Positive residuals are more widespread, showing a general pattern of more phylogenetic diversity than expected given the number of species. The latter can mean that the species to be gained locally are distantly related, or that species that are closely related are being lost. The gain of distant species or loss of redundant ones can be driven by species interactions driving similar species to compete (Webb et al., 2002). As such, studies that quantify and monitor Atlantic Forest species interactions and networks may be particularly relevant under current and future environmental shifts (Sales, Rodrigues & Masiero, 2021). On the other hand, a few areas show consistent trends toward phylogenetic diversity loss relative to species richness (inland São Paulo for Miconieae plants, Boana frogs and Thraupinae birds, a cluster of sites in the northwestern forests for all but the butterflies, and the southern Atlantic Forest for the Miconiae plants and the butterflies). In those areas, the model is predicting the loss of distantly related species, or the gain of closely related ones. This might be pointing at areas where environmental conditions are becoming more extreme and so species persisting, or colonizing, will be more like each other than expected by chance. While the present study is unable to validate these hypotheses, they are testable through field observations and experimental work.

Conclusions and Final Remarks

The modeling framework proposed here allows for the flagging of areas that are at risk of losing diversity in the face of climatic changes. The incorporation of two dimensions of diversity—taxonomic and phylogenetic—enables the identification of areas that might be at risk of losing evolutionary diversity due to environmental change. However, the predictions of the model should be taken carefully when predicting diversity gain, as those seem to be overly optimistic. Because the strength of the predicted change is overestimated by the model, we propose that practitioners be careful when interpreting magnitude; it may be best to focus only on the expected direction of change. This would allow, for example, for a better design of new protected areas and for targeted monitoring of areas either predicted to lose diversity (in general) or to disproportionately lose evolutionary diversity.

Key to this indirect remote sensing effort is the periodicity of the remote sensing information to be correlated with—and used as proxy for—biological diversity. Remote sensing data are available at different periodicities, ranging from days to years, depending on the agencies that collect them, their instruments, and costs. Accessing free data has become easier with the advent of Google Earth Engine (Gorelick et al., 2017), a cloud-based service that not only stores multiple datasets but also allows for free data processing through open code. The framework is easily implemented and can be rerun with every pass of a satellite, incorporating giving us new environmental data and thus adjusting predictions to the real change in near real time. We propose that this use may be further explored as a tool for conservation planning and policy design, using the most up-to-date information.

This framework is also easily adaptable to other groups or areas of interest, by replacing the calibration biodiversity data. It may be applied to other diversity measures and dimensions of biodiversity, although users must carefully assess their degree of correlation with present-day environmental variables. Previous work has shown that metrics that largely reflect historical components or are associated with spatial restriction (e.g., phylogenetic endemism) are not easily predicted by climate descriptors (Paz et al., 2021).

Importantly, our preliminary analyses demonstrate that this framework does not require extensive inventory data for the whole forest, but may be applied to species with curated occurrence data, allowing for indirect spatial monitoring of biodiversity in near real-time. Based on the findings of the present analysis, we propose that it be used only for groups of species that are wide ranging and well sampled in a biome of interest. Because correlations between diversity patterns and environmental correlates are dependent on the area of model training, the model is not to be applied outside the region where it was trained (in this case, the Atlantic Forest biome). Diversity patterns can vary in different geographical areas, making model transfer problematic, especially in regions with area limitations, such as islands.

Supplemental Information

Supplemental Information 1 Supplemental Figures and Tables

Click here for additional data file.

We would like to thank the members of the AF-Biota and NSF Dimensions of Biodiversity projects for discussions that inspired this work. We thank Rob Anderson, Brian Smith, and members of the Carnaval Lab at CUNY for comments on previous versions of this manuscript. We also thank the editor, Shan Kothari, Markus Gastauer and one anonymous reviewer for comments that greatly improved this manuscript.

Additional Information and Declarations

Competing Interests

Author Contributions

Data Availability

The authors declare there are no competing interests.

Andrea Paz conceived and designed the experiments, performed the experiments, analyzed the data, prepared figures and/or tables, authored or reviewed drafts of the article, and approved the final draft.

Thiago S. Silva conceived and designed the experiments, performed the experiments, analyzed the data, authored or reviewed drafts of the article, and approved the final draft.

Ana C. Carnaval conceived and designed the experiments, authored or reviewed drafts of the article, and approved the final draft.

The following information was supplied regarding data availability:

The code is available at GitHub: https://github.com/andrepazv/NearRealTime.

The data used from Paz et al. (2021) is available at Dryad: Paz, Andrea (2021), Environmental correlates of taxonomic and phylogenetic diversity in the Atlantic Forest, Dryad, Dataset, https://doi.org/10.5061/dryad.6m905qfzp.

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
