# Peer review of "A framework for near-real time monitoring of diversity patterns based on indirect remote sensing, with an application in the Brazilian Atlantic rainforest"

_PeerJ, doi:10.7717/peerj.13534_

## Round 0.1 · original submission · Minor Revisions

All reviewers are very enthusiastic about your work and so am I. Please address all comments and clarify your figures:
Figs. 1 and 4 - Include color legend in figure

·

Basic reporting

I am going to copy-paste my entire review into 'Additional Comments' below, but here are a few excerpted comments related to this section:

"I also thought the authors could have done more to bring out what’s interesting about the study specifically in contrast to other studies. Predicting diversity from static climate variables is often good and useful, but wouldn’t really have been a new idea. Predicting diversity from remote sensing products that are being measured continuously does seem more novel, although I’m not too familiar with the literature here. So essentially, I wonder whether this approach could be better put into context within the conceptual space of possible approaches to modeling diversity, highlighting its advantages and disadvantages. For example, it could (as the authors say) enable near-real time monitoring of potential change in diversity, which seems tremendously useful—but it might be harder to use for forecasting into the future, given that projected changes in climate or canopy cover would have to be transformed into the specific variables (e.g. NDVI) the model requires.

The large-scale organization of the manuscript seems sound—for the most part, it was easy to follow. There are a few sections (particularly in the Methods and Results) with which I did struggle, and I hope my comments can help the authors clarify them."

Experimental design

No comments.

Validity of the findings

I am going to copy-paste my entire review into 'Additional Comments' below, but here are a few excerpted comments related to this section:

"I have many rather small comments (see below) but few large comments about the concept or framing of the manuscript. One issue that I thought deserved more attention is touched on at the end of the Discussion (454-484). The core of the issue is: Do we have good reason to think the factors that link climate/NDVI and diversity across space up to the present are the same as the ones that will link climate/NDVI and diversity through time in the future? The authors bring up some related concerns, including the importance of dispersal or establishment lags for explaining why we might not see predicted species gains in empirical data (439-448). But I was surprised not to see the matter brought up earlier, given that I think it’s at the crux of whether the model will be useful in practice. The authors wisely suggest that any ecological conclusions/mechanisms from the model should be tested on the ground (466-484), but I still worry that predictions (especially e.g. of PD vs richness residuals) could easily be overinterpreted, which might be worth forestalling."

Additional comments

Thank you for the chance to learn from and share my thoughts on this manuscript submitted to PeerJ. The manuscript builds on a prior study (Paz et al. 2021 Journal of Biogeography) which builds models to predict diversity (species richness and phylogenetic diversity) of nine clades in the Atlantic Forest of Brazil from remotely sensed NDVI and climatic variables. Here, the authors project the model forward to predict biodiversity change through time. The idea seems really interesting—as the authors point out, it could allow near-real time monitoring of potential changes in biodiversity that might guide on-the-ground research. The authors then ‘ground-truth’ the model on datasets collected by other researchers and show that it has some modest success in predicting actual changes in diversity through time. While the results of this ‘ground-truthing’ are not staggering, the procedure is quite a severe test of the approach, for which the authors should be commended.

I have many rather small comments (see below) but few large comments about the concept or framing of the manuscript. One issue that I thought deserved more attention is touched on at the end of the Discussion (454-484). The core of the issue is: Do we have good reason to think the factors that link climate/NDVI and diversity across space up to the present are the same as the ones that will link climate/NDVI and diversity through time in the future? The authors bring up some related concerns, including the importance of dispersal or establishment lags for explaining why we might not see predicted species gains in empirical data (439-448). But I was surprised not to see the matter brought up earlier, given that I think it’s at the crux of whether the model will be useful in practice. The authors wisely suggest that any ecological conclusions/mechanisms from the model should be tested on the ground (466-484), but I still worry that predictions (especially e.g. of PD vs richness residuals) could easily be overinterpreted, which might be worth forestalling.

I also thought the authors could have done more to bring out what’s interesting about the study specifically in contrast to other studies. Predicting diversity from static climate variables is often good and useful, but wouldn’t really have been a new idea. Predicting diversity from remote sensing products that are being measured continuously does seem more novel, although I’m not too familiar with the literature here. So essentially, I wonder whether this approach could be better put into context within the conceptual space of possible approaches to modeling diversity, highlighting its advantages and disadvantages. For example, it could (as the authors say) enable near-real time monitoring of potential change in diversity, which seems tremendously useful—but it might be harder to use for forecasting into the future, given that projected changes in climate or canopy cover would have to be transformed into the specific variables (e.g. NDVI) the model requires.

The large-scale organization of the manuscript seems sound—for the most part, it was easy to follow. There are a few sections (particularly in the Methods and Results) with which I did struggle, and I hope my comments can help the authors clarify them.

Shan Kothari
shan.kothari@umontreal.ca

Major comments:
76-83: I think this line of reasoning could appear suspect to many readers in light of the case presented Mayfield & Levine (2010 Ecol Lett). In particular, Mayfield & Levine show that competition can also lead to clustering, while (as Cavender-Bares et al. 2009 Ecol Lett point out) environmental filtering can lead to phylogenetic overdispersion if trait evolution is convergent. Any given pattern can thus have multiple plausible explanations in the absence of other context that rules some out. I would also be concerned that the spatial resolution of the data (~10 km) in this study would make it hard to pick up on the signatures of local species interactions (e.g. see Cavender-Bares et al. 2018 American Journal of Botany). This isn’t a problem for the authors’ decision to look at phylogenetic diversity, since we have more reason to care about phylogenetic diversity than just as a guide to local community assembly processes; but it might be more convincing to put some of those other reasons in the foreground instead.
208-241: I realize it’s often hard to describe model calibration and validation procedures in a clear way, but I struggle to understand this section. My understanding is that the static diversity maps were combined with the remote sensing/climate products from the 2002-2014 period, but I’m not sure how the temporal aspect within the 2002-2014 period is handled (or if the predictors were just combined via summary statistics). The paragraph makes continual references to ‘projection through time,’ but I can’t tell whether that refers to something within the 2002-2014 period or to the projections to the 2015-2019 period that are described in the next paragraph. Then follows cross-validation across spatial partitions, which makes sense, but it’s unclear whether the only output from these models used in to build the ‘full’ models for each algorithm is some set of optimal settings, or something else besides. The terms can also be a bit confusing, e.g. line 235 refers to “final tuned models” (presumably for each fold of CV) while line 236 refers to “a full model” (across all data after CV). I think a bit of rewriting could make this section much easier to follow.
225-233: I like the idea behind this spatial partition procedure, but it would be interesting to know what it looks like. How were the regions determined?
288-290: I think this whole ground-truthing approach is commendable, and I really appreciate the thought that went into it. That said—while this comparison to a coin flip (essentially) makes sense, I’m not sure it ends up providing the strongest test of the models’ power. While reading this section, I thought about what my uninformed baseline guess would be about biodiversity change would be—and I decided I would guess that biodiversity is declining in each pixel (here and in most of the world, at large enough spatial scales). Based on the results (367-391), my baseline ‘model’ that every pixel is declining in species richness would outperform the epiphyte model and come quite close to the performance of the amphibian model. This isn’t meant as a criticism, and certainly there must be value to having a more spatially targeted prediction, but I wonder if there’s a way to describe the results here that makes these advantages clearer? I don’t have a specific suggestion, and I don’t mean for this comment to obligate any response, but perhaps it gives something to think about.
331-342: It could be interesting (if not too burdensome) to create similar maps of the change in important predictor variables from 2015-2019 so that we can see by eye how they relate to the predicted outcomes for diversity.
Fig. 2: Based on the caption, does this mean that some pixel(s) have predicted local loss of ~36 Miconieae species between 2015 and 2019? That number seems very large, so I just want to check that I’m understanding correctly.

Minor comments:
Here, I use the → symbol to represent a textual change that may be easier to read. Of course, each of these comments is just a suggestion!

26: “periodical” → “periodic” (periodical most often means a newspaper/magazine)
39: “difficulty to establish” → “difficulty of establishing”
48: “what can be now done” → “what can now be done”
69: “hereon” → “henceforth”
69-72: This sentence is a little ambiguous. The intended meaning, I’m sure, is that species richness and phylogenetic diversity are both correlated with climate and habitat configuration. But I first read it to mean that species richness is correlated with climate, habitat configuration, and phylogenetic diversity. One solution would be to change “also” (70) to “so have.”
101-102: Presumably the studies listed in this parenthetical are now the published studies listed at the end of the sentence? (If so, congratulations!)
103-104: I might recommend taking this clause that begins “but not…” and turning it into a separate sentence so that this sentence coheres a bit better.
108: “machine-learning reproducible framework” → “reproducible machine learning framework”
111-115: Since this list is spelled out in the Methods, it may be fine to cut it here.
112: “Mealastomataceae” → “Melastomataceae”
127-128: “expect that those groups… to be” → “expect that those groups… will be” or “expect those groups… to be”
164-165: It might help to briefly explain what this flexibility constitutes, e.g. is it in general true that alpha hulls around a given set of points will be no larger than the corresponding convex hull? Can they be disjoint?
165: The referent of “they” here is a little unclear. The specific alpha hulls used to create the maps in this (and the prior) study?
166: It’s hard for me to put this note that the datasets “lacked information about the date of observation of each sample” into context. What follows from this fact?
175: What metric of phylogenetic diversity? Faith’s PD?
183: It seems legitimate for the study to claim that its framework would allow near-real time modeling of biodiversity, but I’m not sure it makes sense to describe this set of models demonstrating the framework as “near real-time models.” To me, that term implies that the estimates are being made about the present or very recent past.
184: “the conditions experience” → “the conditions experienced”
188-189: I would write “The two MODIS data products” (or some variant) rather than “Data from MODIS” so that the note on spatial resolution is easier to follow. I might also suggest just writing CHIRPS rather than “the latter.”
195: “NDIV” → “NDVI”
215: “one” → “an” (or vice versa in the previous line)
220: Pluralize “neural network”?
242-245: This sentence was a bit hard to follow and might be better split up into two.
245: “each yearly” → “each year’s” or “each set of yearly”
274: It took me a while to understand exactly what was meant by “had some coverage of the forest.” Perhaps something like “had relatively broad coverage of the region”?
299: Here and in the abstract, the manuscript says that the variables describe change in temperature and precipitation, which seems to leave out NDVI. Perhaps not a big issue, but I would imagine that change in canopy cover as shown by NDVI might explain a major part of the models’ predictive power.
303-304: This clause “few exceptions happened when individual fit was poor ensemble fit was worse” is very hard to understand.
304-306: It might be worth showing some formal statistics if appropriate.
306-307: “The models with good fit for both diversity measures (R2 > 0.5) were observed” → “Models had good fit for both diversity measures”
349: I’m not sure what it means to say “where the Atlantic Dry forests are found up to 2018.”
352: Extra period after “richness.”
360: “as” → “a”?
361: “somehow consistently” → “somewhat consistently”?
383: “has” → “have”
389: Missing word before “an”?
411: “result” → “results”; I’m not really sure what “overestimation of diversity patterns” means.
436: “direction of magnitude of change” → “direction or magnitude of change” or “direction of change”
441: “more difficult documented” → “more difficult to document”
463: Perhaps “suggested by” rather than “enabled by”?
467: “and tested” → “and test them”; “While patterns…, yet the predicted residuals” → “Patterns…, yet the predicted residuals” or “While patterns…, the predicted residuals”

Reviewer 2 ·

Basic reporting

Very good.

Experimental design

Very good.

Validity of the findings

Very good.

Additional comments

Review of MS: A framework for near-real time monitoring of diversity patterns based on indirect remote sensing, with an application in the Brazilian Atlantic rainforest (#70910)
1. The introduction is good and but it should draw parallels with relevant studies from the peer-reviewed literature covering novel concept of essential biodiversity variables and multiple components of biodiversity. Authors may cite the following highly applicable works.

Skidmore, A. K., Coops, N. C., Neinavaz, E., Ali, A., Schaepman, M. E., Paganini, M., ... & Wingate, V. 2021. Priority list of biodiversity metrics to observe from space. Nature ecology & evolution, 5(7), 896-906.

Reddy, C.S. 2021. Remote Sensing of Biodiversity: What to measure and monitor from Space to Species? Biodiversity and Conservation, 30, 2617–2631.

2. The methods section should spell out clearly the multi-source species data, linkage of remote sensing based environmental datasets and validation of results. Towards this, I suggest authors to prepare a flow chart showing methodological framework.

·

Basic reporting

Dear authors, I enjoyed revision and apologize for delays! You link remotely sensed climate data with presence-absence data from nine taxonomic groups and validate with extensive monitoring datasets. The combination of techniques results in a robust framework to estimate alterations in the distribution ranges, and provides an important contribution to enhance biodiversity monitoring. I recommend highly the publication of the manuscript, although some revision is necessary:

Main point: What about cumulative effects of climate? Physiologically, climate severity (e.g., a dry year) may reduce fitness and reproduction of a population, but may be compensated by a following, less severe year, in which the population recovers, while an additional severe year may indeed cause (temporary) local extinction of the population. So I wonder if the consideration of middle-term climatic effects (such as ENSO anomalies) may enhance the predictions.

I would welcome some speculations about underlying mechanisms for mismatches in between observed and modelled data (Figure 4). It seems that matches are concentrated in regions with higher forest cover/less fragmentation, which would indicate higher dispersal abilities for the taxa (although colonization may lag behind environmental alterations as stated in discussion)!

Furthermore, I would welcome Table S2 in the main text, to give the reader a better idea about the model parameters. Consequently, some details about the models may be withdrawn from the result section.

Legend Figure 2: Can slope variation be incorporated in the Figure?

To finish: I am not a native speaker, but some spelling/grammar/editing issues: References in l. 101, Melastomataceae in l. 112, word duplication in l. 298

Experimental design

As stated above: What about cumulative effects of climate? Physiologically, climate severity (e.g., a dry year) may reduce fitness of a population, but may be compensated by a following, less severe year, while an additional severe year may indeed cause (temporary) local extinction of the population. So I wonder if the consideration of middle-term climatic effects (such as ENSO anomalies) may enhance the predictions.

Validity of the findings

OK

Additional comments

Non

---

## Round 0.2 · accepted · Accept

Thank you for addressing the reviewer`s comments. I am happy to accept your manuscript.